# A Long Photoperiod Promoted the Development, Reproduction, and Predation of *Harmonia axyridis* Pallas (Coleoptera: Coccinellidae) at an Average Greenhouse Temperature during the Winter

**DOI:** 10.3390/insects15040214

**Published:** 2024-03-22

**Authors:** Haixia Yu, Xinjuan Yuan, Zhiqiang Xie, Qiqi Zhang, Changying Zheng, Lijuan Sun

**Affiliations:** College of Plant Medicine, Shandong Engineering Research Center for Environment-Friendly Agricultural Pest Management, Qingdao Agricultural University, Qingdao 266109, China; y15634363687@163.com (H.Y.); xinjuany@163.com (X.Y.); xiezhiqiang4396@163.com (Z.X.); qqzhang77@sina.com (Q.Z.); zhengcy67@qau.edu.cn (C.Z.)

**Keywords:** life table, greenhouse, photoperiod, predation, winter

## Abstract

**Simple Summary:**

*Harmonia axyridis* (Pallas) is a widely distributed predatory natural enemy insect in North China but its application is limited in greenhouses owing to a low daily average temperature from December to February of next year in greenhouses and the facultative diapause of adult *H. axyridis*. Owing to the sensitivity of *H. axyridis* to photoperiod, it is of great importance to explore the feasibility of using the photoperiod to regulate the life parameters and predation ability of this ladybird in greenhouses during the winter, but the effects of photoperiods on life parameters and predation by *H. axyridis* are not well understood so far. In this study, development, reproduction, and population parameters were studied by constructing two-sex life tables, and effects of photoperiod on predation were tested by predation experiments in both laboratory and greenhouse settings. The results from this article may provide a basis for utilizing *H. axyridis* in greenhouses during the winter.

**Abstract:**

To explore the feasibility of adjusting the photoperiod to regulate the life parameters and predation ability of *Harmonia axyridis* Pallas in greenhouses during the winter, life tables were constructed for *H. axyridis* under the three following photoperiods: 9L:15D (light/dark), 12L:12D, and 16L:8D at 15 °C, an average greenhouse temperature during the winter when aphids severely damage vegetables. The effects of photoperiods on predation by this ladybird were tested in both laboratory and greenhouse settings. The results showed that increased illumination promoted the development and reproduction of *H. axyridis*; under medium and long photoperiods, the pre-adult periods were 3.61 days and 4.34 days shorter than that under the short photoperiod, respectively, and the fecundity increased by 1.78 and 2.41 times. Population parameters *r*, *λ*, and *R*_0_ increased as illumination time increased, whereas *T* decreased. Increased illumination also increased the predation by third- and fourth-instar larvae and adults. The amounts of predation by fourth-instar larvae and adults increased by 22.16% and 75.09% under the medium photoperiod, and those under the long photoperiod increased by 71.96% and 89.64%, respectively. The numbers of *Myzus persicae* Sulzer predated by *H. axyridis* under the long photoperiod were higher than those under the short photoperiod in a greenhouse, and the predation parameters were influenced.

## 1. Introduction

Members of the family Coccinellidae, which are commonly known as ladybirds, are diurnal in habit and most of their behaviors, such as searching, foraging, and mating, are clearly responsive to light [1,2]. The effects of different photoperiods on development and reproduction have been studied in several species of ladybirds [3,4,5,6,7,8,9], which showed that the photoperiod served as a signal to adjust growth, development, and behavior [10] as well as a response to climate change [11,12]. The photoperiod also affects predation by ladybirds. The amount of predation by *Coccinella septempunctata* L. (Coleoptera: Coccinellidae) and *C. transversalis* Fabricius (Coleoptera: Coccinellidae) [9] larvae and *Cheilomenes sexmaculata* (Coleoptera: Coccinellidae) [13] larvae and adults increased as the illumination increased; however, the effects of different photoperiods on predation by *Harmonia axyridis* Pallas (Coleoptera: Coccinellidae) are not well understood.

*Harmonia axyridis* is a widely distributed and highly adaptive predatory insect [14,15], and it is the dominant predatory ladybird species in North China. *H. axyridis* adults have predation and reproduction problems in unheated and non-illuminated greenhouses during the winter when the temperature is below 20 °C and the illumination time is less than 12 h [16]. In northern China, aphids such as *Myzus persicae* Sulzer (Hemiptera: Aphididae) and *Aphis gossypii* Glover (Hemiptera: Aphididae) cause severe problems in greenhouses from December to February of the following year. During this period, the average temperature in greenhouses is usually about 15 °C and, thus, *H. axyridis* cannot effectively control aphids in greenhouses. The possibility of improving the proliferation capacity and predation ability of *H. axyridis* populations in greenhouses by adjusting illumination time requires further investigation.

Thus, in the present study, we explored the feasibility of using the photoperiod to regulate the life parameters and predation ability of *H. axyridis* in greenhouses during the winter. The temperature was monitored in a greenhouse during the winter to test a daily mean temperature for construction of indoor life tables. Three photoperiods comprising 9L:15D (light/dark), 12L:12D, and 16L:8D were tested to construct life tables for *H. axyridis* at the above mean temperature. The effects of different photoperiods on predation by *H. axyridis* were determined in predation experiments in a laboratory and greenhouse.

## 2. Materials and Methods

### 2.1. Insects

*Harmonia axyridis* eggs were collected from peach trees on the campus at Qingdao Agricultural University (Qingdao, China) in April, 2019. Each *H. axyridis* larva was reared alone in a Petri dish with diameter of 75 mm with *M. persicae* as prey to establish a laboratory colony. The laboratory colony was established at 20 ± 1 °C and 65 ± 10% relative humidity under a photoperiod of 16 h light/8 h dark in an artificial climate box (RGN-300, Ningbo Southeast Instrument Co., Ltd., Ningbo, China).

### 2.2. Assays of Effects of Different Photoperiods on Development and Reproduction

A temperature and humidity recorder (GSP-6, Jiangsu Jingchuang Electric Co., Ltd., Xuzhou, China) was used to monitor the temperature every hour during each day in the greenhouse during December 2021 when aphids were abundant. The average temperature was calculated and was used in the indoor experiment. Eggs collected on the same day were transferred into a Petri dish (diameter = 75 mm) with a piece of moisture filter paper at the bottom. The Petri dish was sealed with plastic film that was punctured with a few very small holes to provide air and placed in an artificial climate box (RDN-300, Ningbo Southeast Instrument Co., Ltd., Ningbo, China). The eggs collected from the laboratory colony on the same day were cultured under three different photoperiods comprising 9L:15D, 12L:12D, and 16L:8D at 15 °C (an average greenhouse temperature) and a relative humidity of 65% ± 10%. After hatching, each larva was picked with a small brush and placed into a new Petri dish for continuous culture with *M. persicae* as prey. Molting and survival by *H. axyridis* were observed and recorded every 24 h. At 7 days after eclosion, males and females were identified under an anatomic microscope and paired, before being placed in a 100 mL volumetric glass beaker, which was covered with plastic film that was punctured with a few very small holes. Eggs were collected every 24 h and the numbers of eggs recorded. If sufficient individuals were not available for pairing, ladybirds were selected from a population cultivated under the same conditions. If an adult died before its mate, it was replaced by one of the same sex and similar age obtained from the population cultivated under the same conditions, but data pertaining to these individuals were excluded from the life table analysis. Eighty eggs were cultured for the assays under each photoperiod.

### 2.3. Assays of Effect of Different Photoperiods on Predation by H. axyridis

#### 2.3.1. Assay of Effects of Different Photoperiods on the Daily Average Amount of Predation

The ladybirds were cultured as described above and specific numbers of third-instar nymphs of *M. persicae* were placed into each Petri dish. The quantity of remaining aphids was checked every 24 h before adding new aphids to maintain a constant quantity. The number of aphids released in the Petri dish was determined based on a preliminary experiment to ensure that the quantity of aphids was adequate. *H. axyridis* pupae were continuously cultured until the eclosion of adults. Each newly emerged adult was fed with a sufficient number of aphids using the method for larvae, and the remaining number of aphids was also counted every 24 h. Adults were paired on the seventh day after eclosion using the same method described above. A sufficient number of aphids were placed in a small beaker for adult culture. At 14, 21, and 28 days after eclosion, the female and male were separated and transferred into a Petri dish (diameter = 5 cm) for separate predation amount assays using the same method applied for larvae and adults before mating. Five females and five males were assayed in each treatment. Three replicates were conducted for each treatment.

#### 2.3.2. Assay of Effects of Different Photoperiods on Predation by *H. axyridis*

*Harmonia axyridis* individuals were cultured from eggs at 9L:15D with *M. persicae* as the prey. Some of the ladybirds were cultured to fourth-instar larvae while the remaining ladybirds were cultured to adults. The larvae were starved for 24 h on the second day of the fourth instar before predation testing and adults on the fifth day after eclosion before predation testing. Predation testing was conducted in a Petri dish with a diameter of 75 mm. The bottom of the Petri dish was covered with a layer of wet filter paper, and 30 wingless *M. persicae* nymphs of third instar (average body weight: about 0.3 mg) were placed in the Petri dish, before releasing a starved ladybird in the Petri dish. The Petri dish was sealed with plastic film, which was then punctured with a few very small holes. Predation tests were conducted in an artificial climate box at 15 °C under photoperiod conditions comprising 9L:15D, 12L:12D, and 16L:8D. The numbers of remaining aphids were recorded after 24 h. Ten ladybirds were tested for each replicate, with three replicates for each treatment. The predation amount under 9L:15D was treated as the control to calculate the changes in the predation rate under medium and long photoperiods.

#### 2.3.3. Assays of Effects of Different Photoperiods on Predation Parameters for *H. axyridis* in Greenhouse

*Harmonia axyridis* was cultured under 9L:15D at 15 °C in an artificial climatic box and 5-day-old adults were used in predation experiments. Third-instar *M. persicae* nymphs with similar body sizes were inoculated with a small brush onto an eggplant that had not been infected previously by pests, and excess aphids were removed after 24 h to obtain prey densities of 5, 10, 20, 50, 80, and 110 per plant. An eggplant seedling with *M. persicae* was carefully placed in a cage with dimensions of 28 cm × 28 cm × 40 cm, before releasing an adult ladybird that had been starved for 24 h into the cage. Experiments were conducted simultaneously under photoperiods of 9L:15D, 12L:12D, and 16L:8D in the greenhouse. The ladybird was removed from the cage after 24 h and the number of *M. persicae* remaining on the eggplant seedlings was counted. The temperature and humidity in the greenhouse were recorded during the experiment using a GSP temperature and humidity recorder. Three replicates were conducted for each prey density, and each predation experiment was conducted three times.

### 2.4. Statistical Analysis

TWOSEX-MSChart software [17,18,19] was used to analyze the biological parameters for the *H. axyridis* population, including the development duration for each stage, adult pre-oviposition period comprising the duration from eclosion of a female to first oviposition, total pre-oviposition period comprising the duration from a female egg to first oviposition, age-instar survival rate (*s_xj_*), age-specific survival rate (*l_x_*), female fecundity (*f_x_*), age-specific fecundity (*m_x_*), age-specific reproductive survival rate (*l_x_m_x_*), intrinsic growth rate (*r*), finite growth rate (*λ*), net fecundity (*R*_0_), and average generation period (*T*). The mean life table parameters and standard estimations were calculated using the bootstrap method [20]. The standard errors for all above life table parameters were estimated using the bootstrap procedure with 100,000 iterations. To assess the differences among the various treatments, the paired bootstrap test was employed, considering the confidence interval of differences [21]. The formulae used for calculating life table parameters with TWO-SEX-MSChart software are as follows.
$$l_{x}=\sum_{j=1}^{k}s_{xj},$$
where *j* indicates the development stage, *k* is the number of stages.
$$m_{x}=\sum_{j=1}^{k}s_{xj}f_{xj}/\sum_{j=1}^{k}s_{xj},$$
where $f_{xj}$ indicates the fecundity of a female adult of age *x* and stage *j*.
$$e_{xj}=\sum_{i=x}^{\infty }\sum_{y=j}^{k}s_{iy}^{\prime },$$
where $s_{iy}^{\prime }$ indicates the probability of an individual of age *x* and stage *j* surviving to age *i* and stage *y*.
$$v_{xj}=\frac{e^{r(x+1)}}{s_{iy}^{\prime }}\sum_{i=x}^{\infty }e^{-r(i+1)}\sum_{y=j}^{k}{s\prime }_{iy}f_{iy},$$
where *r* indicates the intrinsic population growth rate, *f_iy_* indicates the fecundity of a female adult of age *i* and stage *y*, and $\sum_{x=0}^{\infty }e^{-r(x+1)}$*l_x_m_x_* = 1 is used to calculate *R*_0_.
$$R_{0}=\sum_{x=0}^{\infty }l_{x}m_{x},$$
where *l_x_* is the age-specific survival rate and *m_x_* is the fertility at a specific age.
$$T=(\ln R_{0})/r$$
$$\lambda =e^{r}$$

The average daily predation amount of each instar by larvae was calculated by using Excel 2010 software according to the following formula: E = total predation amount in an instar/duration (day). The increase in the rate of predation was calculated by using the following formula: Increase rate (%) = (average predation amount under treatment−average predation amount under control)/average predation amount under control × 100. The predation functional response of *H. axyridis* to *M. persicae* was fitted by using the Holling-II disk equation. The Holling-II equation comprises *Na* = *a × NoT*/(1 + *a × ThNo*), where *a* is the instantaneous attack rate, *Th* is the handling time, *Na* is the daily predation amount, *No* is the prey density, *T* is the time duration of the test, which was 1 day in our experiment, 1/*Th* is the theoretical maximum daily predation amount, and *a*/*Th* is the maximum daily predation amount. The chi-square test was used to determine whether the equation conformed to the Holling-II model [22]. The model proposed by Holling comprising *S* = *A*/(1 + *A × Th × No*), which reflects the relationship between search efficiency (*S*) and prey density, was used to calculate the search efficiency of *H. axyridis* on *M. persicae* under different prey densities, where *A* = *a*. Differences between predation amounts of *H. axyridis* under different photoperiods and that between different developmental stages under the same photoperiod were statistically analyzed by one-way ANOVA, LSD, at *p* = 0.05 with SPSS 27.0 (IBM, Armonk, NY, USA). The interactions of photoperiod and developmental stage were analyzed by two-factor analysis, LSD, at *p* = 0.05 with SPSS 27.0.

## 3. Results

### 3.1. Daily Average Temperature in the Greenhouse during December 2021

Figure 1 shows that the daily average temperature in the greenhouse during December 2021 was about 15.0 °C (calculated according to daily average temperature), where the highest daily average temperature was 20.5 °C and the lowest daily average temperature was 6.7 °C.

### 3.2. Development and Reproduction Parameters for H. axyridis under Different Photoperiods

The life table for *H. axyridis* was constructed at 15 °C according to the results in Section 3.1. Table 1 shows that increasing the illumination period at 15 °C accelerated the rates of development and maturation for *H. axyridis*. Under medium and long photoperiods, the pre-adult period was shortened by 3.61 days and 4.34 days, respectively, compared with that under the short photoperiod (df = 2, *p* < 0.05). Under medium and long photoperiods, the pre-oviposition period was shortened by 14.51 days and 29.58 days, respectively, compared with that under the short photoperiod (df = 2, *p* < 0.05). The fecundity of all female *H. axyridis* (*F*) also increased with the illumination time, where the fecundities under medium and long photoperiods were 1.78 times and 2.41 times higher, respectively, than that under the short photoperiod (df = 2, *p* < 0.05). The proportion of ovipositing females under the short photoperiod was 96.30%, which did not differ from that under medium and long photoperiods (both were100%) (df = 2, *p >* 0.05). The fecundity of ovipositing females (*Fr*) under the short photoperiod was 229.0 per female, which also differed from that under medium and short photoperiods (df = 2, *p <* 0.05). The pupal duration was not affected by the illumination time. The longevities of females and males decreased as the illumination time decreased, and they differed under different photoperiods (df = 2, *p* < 0.05).

### 3.3. H. axyridis Population Parameters under Different Photoperiods

Life table parameters were calculated for *H. axyridis* populations under different photoperiods by using TWOSEX-MSChart life table software, i.e., *r*, *λ*, *R*_0_ (number of offspring produced by one female), and *T*. Table 2 shows that the population parameters comprising *r*, *λ*, and *R*_0_ all increased with the illumination time, and they differed among photoperiods (df = 2, *p* < 0.05). The effects of different photoperiods on *T* differed from those on the other three parameters, where *T* decreased as the illumination time increased, with significant differences among photoperiods (df = 2, *p* < 0.05).

### 3.4. Daily Predation Amounts by H. axyridis under Different Photoperiods

Figure 2 shows that the daily predation amounts by first- and second-instar larvae did not differ among photoperiods. By contrast, the daily predation amounts by third- and fourth-instar larvae increased as the illumination time increased. The average daily predation amount by third-instar larvae was 24.0 under the long photoperiod, which differed from those under the short photoperiod (15.1) (F = 47.622, df = 2, 2, *p* < 0.001) and medium photoperiod (17.2) (F = 47.622, df = 2, 2, *p* < 0.001), while the average daily predation amount under 9L:15D did not differ from that under 12L:12D (F = 47.622, df = 2, 2, *p* = 0.074). The average daily predation amount by fourth-instar larvae was 29.2 under the long photoperiod, which differed from those under the short photoperiod (22.0) (F = 57.672, df = 2, 2, *p* < 0.001) and medium photoperiod (25.4) (F = 57.672, df = 2, 2, *p* = 0.001), and the daily predation amount by fourth-instar larvae also differed significantly between short and medium photoperiods (F = 57.672, df = 2, 2, *p* = 0.003). And there is a significant interaction between photoperiod and developmental stages (F = 13.715, df = 11, 24, *p* < 0.001).

Figure 3 shows that photoperiod also affected the daily predation amount by *H. axyridis* adults. The daily predation amounts by adults from the first day to the fourth day after eclosion did not differ between photoperiods. However, the predation amount by adults tended to increase as the illumination time increased from the fifth day after eclosion, and the daily predation amounts at the fifth day after eclosion differed between the long photoperiod and the short photoperiod (F = 26.446, df = 2, 2, *p* = 0.001) and also differed between the medium photoperiod and the short photoperiod (F = 26.446, df = 2, 2, *p* = 0.001) but not differ between the medium photoperiod and the long photoperiod (F = 26.446, df = 2, 2, *p* = 0.436). The daily predation amounts by adults differed between the three photoperiods from the sixth day after eclosion (6-day-old adults: long photoperiod–medium photoperiod: F = 92.453, df = 2, 2, *p* = 0.016; long photoperiod–short photoperiod: F = 92.453, df = 2, 2, *p* < 0.001; medium photoperiod–short photoperiod: F = 92.453, df = 2, 2, *p* < 0.001; 7-day-old adults: long photoperiod–medium photoperiod: F = 76.436, df = 2, 2, *p* = 0.002; long photoperiod–short photoperiod: F = 76.436, df = 2, 2, *p* < 0.001; medium photoperiod–short photoperiod: F = 76.436, df = 2, 2, *p* < 0.001; 14-day-old adults: long photoperiod–medium photoperiod: F = 26.717, df = 2, 2, *p* = 0.014; long photoperiod–short photoperiod: F = 26.717, df = 2, 2, *p* < 0.001; medium photoperiod–short photoperiod: F = 26.717, df = 2, 2, *p* = 0.008; 21-day-old adults: long photoperiod–medium photoperiod: F = 36.404, df = 2, 2, *p* = 0.003; long photoperiod–short photoperiod: F = 36.404, df = 2, 2, *p* < 0.001; medium photoperiod–short photoperiod: F = 36.404, df = 2, 2, *p* = 0.010; 28-day-old adults: long photoperiod–medium photoperiod: F = 24.162, df = 2, 2, *p* = 0.013; long photoperiod–short photoperiod: F = 24.162, df = 2, 2, *p* < 0.001; medium photoperiod–short photoperiod: F = 24.162, df = 2, 2, *p* = 0.013). It also can be seen from Figure 3 that the predation amount by *H. axyridis* increased significantly with aging of the adults under each photoperiod (long photoperiod: F = 214.546, df = 9, 20, *p* < 0.001; medium photoperiod: F = 62.510, df = 9, 20, *p* < 0.001; short photoperiod: F = 66.877, df = 9, 20, *p* < 0.001). And there is a significant interaction between photoperiod and instar (F = 13.459, df = 29, 60, *p* < 0.001).

### 3.5. Regulatory Effect of Photoperiod on Predation by H. axyridis

Figure 4 shows that increasing the illumination time increased the daily predation amounts by *H. axyridis* larvae and adults. When fourth-instar *H. axyridis* larvae reared under the short photoperiod were transferred to the medium photoperiod, the predation amounts increased by 22.16%; when transferred to the long photoperiod, the predation amounts increased by 75.09%, and the increase rate under the medium photoperiod differed from that under the long photoperiod (F = 579.342, df = 1, 1, *p* < 0.001). In addition, when 5-day-old *H. axyridis* adults reared under the short photoperiod were transferred to the medium photoperiod and the long photoperiod, the predation amounts increased by 71.78% and 89.64%, respectively (F = 579.342, df = 1, 1, *p* < 0.001). The increase rate for predation by fourth-instar larvae and adults differed under the same photoperiods (medium photoperiod: F = 457.350, df = 1, 1, *p* < 0.001; long photoperiod: F = 457.350, df = 1, 1, *p* < 0.001). And there is a significant interaction between photoperiod and developmental stages (F = 132.944, df = 3, 8, *p* < 0.001).

### 3.6. Effects of Photoperiod on Predation Parameters of H. axyridis

The predation functional reactions by *H. axyridis* under three different photoperiods all conformed to the Holling-II equation in the greenhouse, and the temperature and humidity during the experiment in the greenhouse are shown in Figure 5. The square of correlation coefficients (*R*^2^) between the prey density and daily predation amounts were all higher than 0.90, and chi-square test results showed that the fitted equation reflected the density-dependent effect on predation by *H. axyridis* (Table 3).

Table 4 shows that the search efficiency by *H. axyridis* adults for *M. persicae* decreased as the aphid density increased under each photoperiod, and the photoperiod significantly affected the search efficiency by *H. axyridis* adults. When the prey density was 5 and 10 aphids per plant, the search efficiencies by *H. axyridis* adults for *M. persicae* differed between the medium photoperiod and the long photoperiod (5 per plant: F = 7.817, df = 3, 3, *p* = 0.010; 10 per plant: F = 4.869, df = 3, 3, *p* = 0.022) but did not differ between the short photoperiod and the long photoperiod (5 aphids per plant: F = 7.817, df = 3, 3, *p* = 0.022; 10 aphids per plant: F = 4.869, df = 3, 3, *p* = 0.333) and the between short photoperiod and the medium photoperiod (5 aphids per plant: F = 7.817, df = 3, 3, *p* = 0.547; 10 aphids per plant: F = 4.869, df = 3, 3, *p* = 0.090). When the prey density was 20 and 50 aphids per plant, the search efficiencies by *H. axyridis* adults for *M. persicae* not only differed between the short photoperiod and the long photoperiod (20 aphids per plant: F = 25.571, df = 3, 3, *p* = 0.003; 50 aphids per plant: F = 16.498, df = 3, 3, *p* = 0.001) but also differed between the short photoperiod and the medium photoperiod (20 aphids per plant: F = 25.571, df = 3, 3, *p* < 0.001; 50 aphids per plant: F = 16.498, df = 3, 3, *p* = 0.006), but no significance was found between the medium photoperiod and the long photoperiod (20 aphids per plant: F = 25.571, df = 3, 3, *p* = 0.056; 50 aphids per plant: F = 16.498, df = 3, 3, *p* = 0.217). When the prey density was 80 and 110 aphids per plant, the search efficiency differed between the three photoperiods (80 aphids per plant: long photoperiod–short photoperiod: F = 45.152, df = 3, 3, *p* < 0.001; long photoperiod–medium photoperiod: F = 45.152, df = 3, 3, *p =* 0.018; medium photoperiod–short photoperiod: F = 45.152, df = 3, 3, *p =* 0.001; 110 aphids per plant: long photoperiod–short photoperiod: F = 47.734, df = 3, 3, *p* < 0.001; long photoperiod–medium photoperiod: F = 47.734, df = 3, 3, *p =* 0.010; medium photoperiod–short photoperiod: F = 47.734, df = 3, 3, *p =* 0.001).

Table 5 shows that the instantaneous attack rate and handling time on *M. persicae* by *H. axyridis* decreased as the illumination time increased. The instantaneous attack rate on *M. persicae* by *H. axyridis* not only differed between the long photoperiod and the short photoperiod (F = 12.045, df = 3, 3, *p* = 0.003) but also differed between the medium photoperiod and the long photoperiod (F = 12.045, df = 3, 3, *p* = 0.009) but did not differ between the medium photoperiod and the short photoperiod (F = 12.045, df = 3, 3, *p* = 0.155). The handling time on *M. persicae* by *H. axyridis* differed between the three photoperiods (long photoperiod–short photoperiod: F = 20.834, df = 3, 3, *p* = 0.001; long photoperiod–medium photoperiod: F = 20.834, df = 3, 3, *p =* 0.041, medium photoperiod–short photoperiod: F = 20.834, df = 3, 3, *p =* 0.009).

### 3.7. Maximum Daily Predation Amounts by H. axyridis on M. persicae in the Greenhouse under Different Photoperiods

The maximum daily predation amount is the maximum theoretical predation amount of a ladybug in a day, which is equal to the time unit 1 (day) divided by the time (day) needed by a ladybug to handle one aphid. The maximum daily predation amounts by *H. axyridis* on *M. persicae* under different photoperiods are shown in Figure 6. The maximum daily predation amount by *H. axyridis* adults was highest under the long photoperiod at 78.1, and it differed from that under the short photoperiod (F = 11.109, df = 3, 3, *p* = 0.004). The maximum daily predation amount by *H. axyridis* adults under the medium photoperiod (73.9) was significantly higher than that under the short photoperiod (56.1) (F = 11.109, df = 3, 3, *p =* 0.011), but the maximum daily predation amounts by *H. axyridis* adults did not differ between the medium photoperiod and the long photoperiod (F = 11.109, df = 3, 3, *p =* 0.432).

## 4. Discussion

In the present study, we found that increasing the illumination time to create a long photoperiod significantly accelerated the development of *H. axyridis*, improved its fecundity, increased the population growth rate and finite rate of increase, and enhanced the predation ability of *H. axyridis* under the average greenhouse temperature during the winter, thereby indicating that increasing the illumination time was conducive to the proliferation of the *H. axyridis* population under a low temperature. However, Reznik et al. [6] and Qin et al. [16] obtained different results where they showed that illumination for less than 14 h at 20 °C accelerated the development of *H. axyridis*. These different results indicate that *H. axyridis* may respond differently to various photoperiods under different temperature conditions because the temperature affects the photoperiod response in insects [23,24,25]. In addition, these different results may be explained by the different ladybird populations investigated. 

It was reported that the incidence of diapause of *H. axyridis* adults under the short photoperiod was rather high [16,25,26]; however, only two females were found to have performed no oviposition under the short photoperiod in our study. We found it was difficult to judge if it would enter diapause because they did not survive to the oviposition period. However, we found an obvious delay of the pre-oviposition period under the short photoperiod. The delayed pre-oviposition period under a short photoperiod should be one of the symbols of diapause. The proportion of oviposition females under the short photoperiod was found relatively high in our study; this makes us suppose that adequate food resources may alleviate the induction of diapause by a short photoperiod.

In a previous study, *H. axyridis* adults were sensitive to the photoperiod at 1–4 days after eclosion when the photoperiod induced diapause [26]. In the present study, we found that the predation amount by *H. axyridis* adults aged 1–4 days was not affected by the photoperiod, but differences in the predation amount were found at the fifth day after eclosion. The reason for low predation during the photoreception period in adults requires further research. Abrupt decreases in food intake and arrested development of the reproductive organs are consistent features of reproductive diapause in insects [27,28,29,30,31]. Therefore, the change in predation by *H. axyridis* induced by the illumination time may have been related to diapause but the specific mechanism involved needs further study. We also found that the effect of increasing illumination on the predation ability of *H. axyridis* was higher in the adults than in larvae and, thus, it is preferable to release *H. axyridis* adults to control aphids in greenhouses, and controlling the photoperiod could improve the biological control efficiency of *H. axyridis*.

The effects of different photoperiods on predation by *H. axyridis* have generally been evaluated based on the predation amount [9,13,32]. In the present study, predation tests conducted indoors and in a greenhouse showed that increasing the illumination time increased the daily predation amount by larvae and adults, and changing the illumination time modified the predation parameters for *H. axyridis* at a temperature of 15 °C. Increasing the illumination time reduced the instantaneous attack rate on prey by *H. axyridis* but it significantly enhanced the search efficiency of *H. axyridis* under most prey densities and shortened the handling time, thereby contributing to the greater predation ability of *H. axyridis* under a long photoperiod.

In winter, when light is insufficient, increasing illumination time promoted the growth, flowering, and fruit setting of crops [33,34]. In North China, supplementary lighting is a common practice for increasing vegetable production in greenhouses and it allows the possibility of modifying the photoperiod to improve the predation efficiency of *H. axyridis* on aphids in greenhouses. In the present study, we studied the effects of different photoperiods on development, reproduction, and predation by *H. axyridis* under the average temperature in a greenhouse during the winter and, thus, our findings may provide the basis for utilizing *H. axyridis* more effectively in greenhouses during the winter.

## Figures and Tables

**Figure 1 insects-15-00214-f001:**
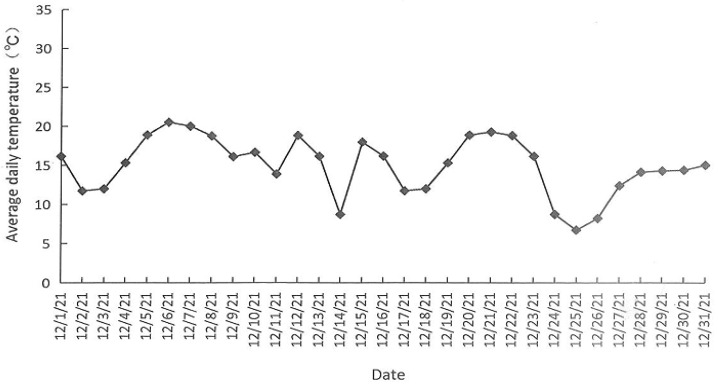
Daily average temperature in the greenhouse during December 2021.

**Figure 2 insects-15-00214-f002:**
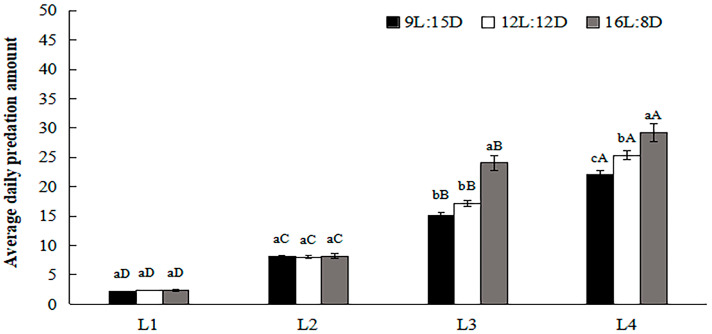
Average daily predation amount by each larval instar of *H. axyridis* to *Myzus persicae* Sulzer under different photoperiods. Data in the figure represent the mean ± SD. L_1_, L_2_, L_3_, and L_4_ represent first-instar larvae, second-instar larvae, third-instar larvae, and fourth-instar larvae, respectively. Different lowercase letters on the columns indicate a significant difference between amounts of prey consumed by same-instar larvae under different light treatments at *p* < 0.05 according to LSD by one-way ANOVA using SPSS 27.0. Different uppercase letters on the columns indicate a significant difference between amounts of prey consumed by *H. axyridis* of different instars under the same photoperiod at *p* < 0.05 according to LSD by one-way ANOVA using SPSS 27.0.

**Figure 3 insects-15-00214-f003:**
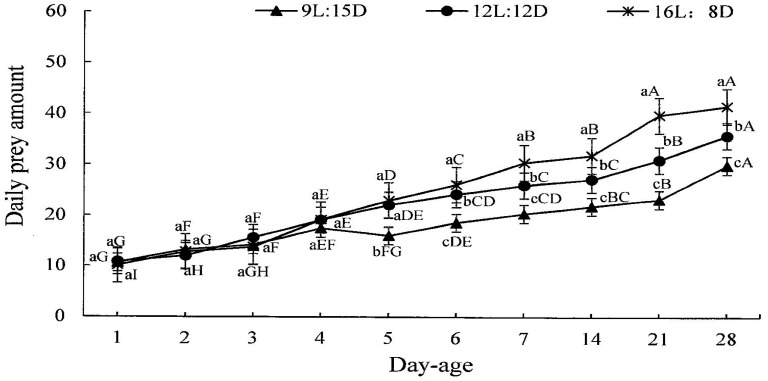
Average daily predation amounts by *H. axyridis* adults to *M. persicae* at different days after eclosion under different photoperiods. Data in the figure represent the mean ± SD. Different lowercase letters in the figure indicate a significant difference between amounts of prey consumed by the same-day-aged adult under different photoperiods according to LSD at *p* < 0.05 by one-way ANOVA using SPSS 27.0. Different uppercase letters in the figure indicate a significant difference between amounts of prey consumed by adults of different day age under the same photoperiod at *p* < 0.05 according to LSD by one-way ANOVA using SPSS 27.0.

**Figure 4 insects-15-00214-f004:**
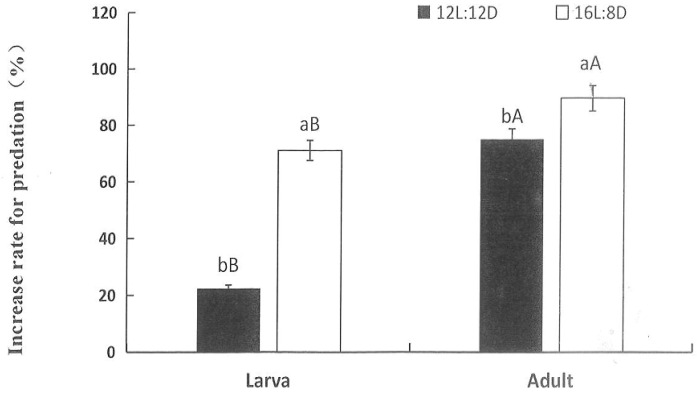
The increase rate for predation of fourth-instar larvae and 5-day-old adults of *H. axyridis* to *M. persicae* under different photoperiods. Different lowercase letters indicate a significant difference between the increase rate for predation of 5-day-old adults or 4th-instar larvae under medium photoperiod and long photoperiod according to two-factor analysis, LSD, at *p* < 0.05 using SPSS 27.0. Different uppercase letters on the columns indicate a significant difference between the increase rate for predation of 4th-instar larvae and 5-day-old adults under the same photoperiod according to two-factor analysis, LSD, at *p* < 0.05 using SPSS 27.0.

**Figure 5 insects-15-00214-f005:**
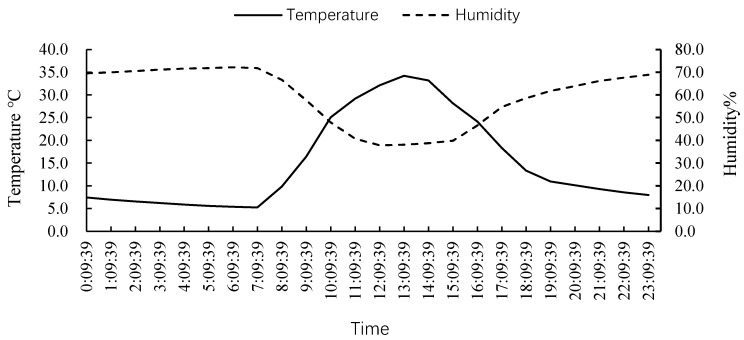
Temperature and humidity in the greenhouse on the day when predation experiments were performed.

**Figure 6 insects-15-00214-f006:**
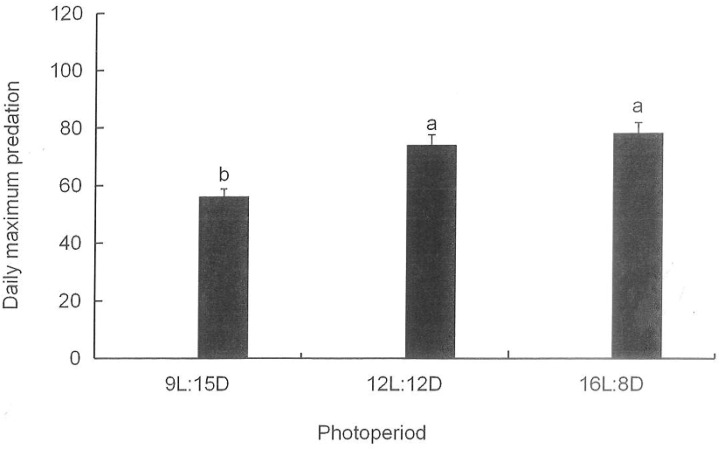
Maximum daily predation amounts by adult *H. axyridis* to *M. persicae* under different photoperiods. The maximum daily predation amount is the maximum theoretical predation amount of a ladybug in a day, which is equal to the time unit 1 (day) divided by the time (day) needed by a ladybug to handle one aphid. Different lowercase letters on columns indicate a significant difference between the daily maximum predation amounts under different photoperiods according to LSD at *p* < 0.05 by one-way ANOVA using SPSS 27.0.

**Table 1 insects-15-00214-t001:** Development and reproduction parameters for *Harmonia axyridis* Pallas under different photoperiods.

Parameters	9L:15D	12L:12D	16L:8D
Duration of egg/d	9.31 ± 0.06 a	9.29 ± 0.08 a	9.34 ± 0.08 a
Duration of 1st instar/d	5.36 ± 0.17 a	5.31 ± 0.18 a	5.71 ± 0.24 a
Duration of 2nd instar/d	4.97 ± 0.14 b	5.33 ± 0.17 a	4.39 ± 0.13 c
Duration of 3rd instar/d	5.10 ± 0.17 b	5.57 ± 0.24 a	4.80 ± 0.12 c
Duration of 4th instar/d	10.81 ± 0.20 a	8.68 ± 0.20 b	9.00 ± 0.17 b
Duration of larva/d	26.24 ± 0.26 a	24.77 ± 0.26 b	23.73 ± 0.29 c
Duration of pupa/d	15.61 ± 0.17 a	15.65 ± 0.27 a	15.62 ± 0.21 a
Pre-adult period/d	52.63 ± 0.26 a	49.02 ± 0.33 b	48.29 ± 0.21 c
Longevity of female/d	162.83 ± 4.57 a	146.62 ± 6.89 b	136.03 ± 7.42 c
Longevity of male/d	169.33 ± 4.18 a	148.84 ± 3.69 b	138.17 ± 3.32 c
Proportion of ovipositing females/%	96.30 ± 2.14 a	100.00 ± 0.00 a	100.00 ± 0.00 a
Fecundity of ovipositing females (*Fr*)	229.0 ± 6.2 c	380.4 ± 12.6 b	513.1 ± 28.4 a
Fecundity of all females (*F*)	213.2 ± 22.4 c	380.4 ± 12.6 b	513.1 ± 28.4 a
Pre-oviposition period (APOP)/d	53.37 ± 5.53 a	38.86 ± 1.87 b	23.79 ± 0.64 c
Total pre-oviposition period (TPOP)/d	104.96 ± 5.20 a	87.27 ± 1.94 b	73.03 ± 4.10 c

Standard errors were estimated using 100,000 bootstrap resampling iterations. The paired bootstrap test was used to detect the difference between different photoperiods. Significant differences between different treatments of the same parameter are indicated by a, b, and c (*p* < 0.05).

**Table 2 insects-15-00214-t002:** Population life table parameters for *H. axyridis* under different photoperiods.

Population Parameter	9L:15D	12L:12D	16L:8D
Intrinsic rate of increase (*r*, per day)	0.0329 ± 0.0013 c	0.0452 ± 0.0016 b	0.0556 ± 0.0018 a
Finite rate of increase (*λ*, per day)	1.0335 ± 0.0014 c	1.0463 ± 0.0017 b	1.0571 ± 0.0019 a
Net reproduction rate (*R*_0_)	76.37 ± 12.21 c	137.89 ± 20.91 b	185.89 ± 26.65 a
Mean generation time *T* (days)	131.40 ± 1.36 a	108.69 ± 1.37 b	93.88 ± 0.85 c

Data in the table represent the mean ± SE. The different letters after the peer data indicate a significant difference (*p* < 0.05) between the data points, as revealed in the paired bootstrap test.

**Table 3 insects-15-00214-t003:** Predation functional reaction equations for *H. axyridis* to *M. persicae* under different photoperiods.

Photoperiod	Replicates	Functional Response Equation	*R* ^2^	*χ* ^2^
9L:15D	1	*Na* = 1.1175 × *No*/(1 + 1.1175 × 0.0162*No*)	0.9948	0.13
	2	*Na* = 1.1056 × *No*/(1 + 1.1056 × 0.0200*No*)	0.9866	0.98
	3	*Na* = 1.0564 × *No*/(1 + 1.0564 × 0.0161*No*)	0.9965	0.63
12L:12D	1	*Na* = 1.0438 × *No*/(1 + 1.0438 × 0.0114*No*)	0.9980	0.43
	2	*Na* = 1.0455 × *No*/(1 + 1.0455 × 0.0136*No*)	0.9906	0.81
	3	*Na* = 1.0884 × *No*/(1 + 1.0884 × 0.0134*No*)	0.9975	0.08
16L:8D	1	*Na* = 0.9743 × *No*/(1 + 0.9743 × 0.0099*No*)	0.9924	0.70
	2	*Na* = 1.0064 × *No*/(1 + 1.0064 × 0.0099*No*)	0.9926	0.71
	3	*Na* = 0.9968 × *No*/(1 + 0.9968 × 0.0092*No*)	0.9955	0.65

The functional response equation in the table is expressed by Holling-II disk: *Na = a* × *NoT*/(1 + *a* × *ThNo*), where *a* is the instantaneous attack rate, *Th* is the handling time, *Na* is the daily predation amount, *No* is the prey density, *T* is the time duration of the test, which was 1 day in our experiment, 1/*Th* is the theoretical maximum daily predation amount, and *a*/*Th* is the maximum daily predation amount.

**Table 4 insects-15-00214-t004:** The search efficiency by *H. axyridis* for *M. persicae* under different photoperiods.

Prey Density (Aphids/Plant)	9L:15D	12L:12D	16L:8D
5	0.9980 ± 0.0257 ab	1.0090 ± 0.0216 a	0.9456 ± 0.0140 b
10	0.9183 ± 0.0244 ab	0.9477 ± 0.0138 a	0.9030 ± 0.0129 b
20	0.7920 ± 0.0273 b	0.8451 ± 0.0036 a	0.8284 ± 0.0132 a
50	0.5612 ± 0.0323 b	0.6383 ± 0.0143 a	0.6639 ± 0.0177 a
80	0.4348 ± 0.0311 c	0.5130 ± 0.0191 b	0.5541 ± 0.0198 a
110	0.3549 ± 0.0286 c	0.4289 ± 0.0204 b	0.4754 ± 0.0202 a

Data in the table represent the mean ± SD (*n* = 3). Different lowercase letters in the same row indicate a significant difference between the search efficiencies by *H. axyridis* for the same prey density under different photoperiods according to LSD at *p* < 0.05 by one-way ANOVA using SPSS 27.0.

**Table 5 insects-15-00214-t005:** The instantaneous attack rate by *H. axyridis* on *M. persicae* and handling time under different photoperiods.

Photoperiod	Instantaneous Attack Rate (*A*)	Handling Time (*Th*/*d*)
9L:15D	1.0931 ± 0.0324 a	0.0174 ± 0.0022 a
12L:12D	1.0592 ± 0.0252 a	0.0128 ± 0.0012 b
16L:8D	0.9925 ± 0.0165 b	0.0097 ± 0.0004 c

Data in the table represent the mean ± SD. Different lowercase letters in the same column indicate a significant difference between the instantaneous attack rates by *H. axyridis* on prey or the handling time under different photoperiods according to LSD at *p* < 0.05 by one-way ANOVA using SPSS 27.0.

## Data Availability

The data presented in this study are available upon request from the corresponding author.

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
