# Peer review of "A Long Photoperiod Promoted the Development, Reproduction, and Predation of Harmonia axyridis Pallas (Coleoptera: Coccinellidae) at an Average Greenhouse Temperature during the Winter"

_insects, 2024, doi:10.3390/insects15040214_

Round 1
Reviewer 1 Report
Comments and Suggestions for Authors
The authors investigated the influence of photoperiod on prey consumption by larvae and adults, the rate of development, fecundity, and some other biological parameters of a biocontrol agent, predatory ladybird Harmonia axyridis. Although photoperiodic responses of this insect have been investigated by several researchers, the results of the earlier studies are rather different suggesting, in particular, the difference between geographic populations. The influence of day length on the rate of predation was not earlier investigated. In addition, the earlier experiments were conducted under laboratory conditions, at constant temperatures, whereas the present study includes experiments performed in greenhouses, under daily thermal rhythms. Therefore the results obtained by the authors can be particularly interesting for the application of biocontrol of insect pests and the manuscript can be published, although before publication some improvements and corrections should be made (see below).
Major comments
It is well known that short photoperiods can induce reproductive diapause in adults of some ladybirds and, in particular, in H. axyridis (citations [1, 2, 4, 6, 16 etc.] in this manuscript). Fecundity of diapausing females is zero. That is why the distribution of the data on fecundity of females reared under the short day conditions is usually bimodal and the results should be better presented not by one parameter (mean fecundity) but by two parameters: (1) the proportion of ovipositing (non-diapausing) females and (2) mean fecundity of ovipositing (non-diapausing) females. However, only the means for all (both ovipositing and diapausing) females is given in Table 1. From these data it is seen that SE under the short day conditions was rather high (more than 10% of the mean) whereas under the 12 and 16 h photoperiods it was about 5% of the mean. This difference suggests that the distribution of fecundity data was possibly bimodal. Hence it is possible that in this study some females also entered diapause. If yes, the data on the proportion of ovipositing (non-diapausing) females and on the mean fecundity of ovipositing females should be given as two different parameters. Generally, it is a bit strange that facultative winter diapause was mentioned in the summary (line 14) but was just ignored in the study. Besides, the presence of some females with zero fecundity means that the statistical distribution of the total data for the whole sample was very far from being normal. Therefore, using of parametric statistics is not correct in this case. On the other hand, in the case if females with zero fecundity were absent in the present study (which would mean that diapause was not induced) this should be clearly stated in the Results and at least briefly mentioned in the Discussion (in comparison with many other studies where the incidence of diapause under the short day conditions was rather high).
In addition, it is also known that diapause can be manifested also as the long-term delay of oviposition. Therefore, the above comments are valid for the preovipoisition period.
Minor comments and corrections
Lines 11, 23, 50, 55, 212, etc.: short Latin name of the aphid is not needed after the full name.
Lines 51, 71: In the beginning of a sentence the full (not abbreviated) Latin genera name should be used.
Line123: Please, describe the prey used in this experiment in more details: “the same size” is important but the size itself (e.g. as the mean weight in mg) and stage (larvae or adults) should be also indicated for the comparison of your data with other similar studies.
Line 214: Why in Table 1 SE (standard errors) are given whereas in all other tables and figures SD are shown (lines 246 etc.). I would suggest using SD everywhere as it does not depend on sample size and therefore it is more reliable measure of variation.
Lines 246-253: I guess this text is a part of the legend to Fig. 2. If yes, it should be properly formatted and replaced.
Line 254: Delete ‘the’ before ‘photoperiod’.
Lines 285-290: I guess this text is a part of the legend to Fig. 3. If yes, it should be properly formatted and replaced.
Lines 308-313: I guess this text is a part of the legend to Fig. 4. If yes, it should be properly formatted and replaced
Table 3 – The indication of photoperiod includes replicate serial number and thus differs from that in text and other tables and figures. This is confusing. Please, use the same indication over the whole manuscript and replace replicate numbers in the next column.
Line 357 (heading of Table 4): “The” should not be in the bold font; “Search” should not be capitalized.
Line 373: I guess not “in the same row” but “in the same column”.
Lines 461 and 497: references [6] and [23] are one and the same publication.
Reviewer 2 Report
Comments and Suggestions for Authors
In this study, the authors measured the development, reproduction, and population parameters by constructing two-sex life tables, and effect of photoperiod on predation of Harmonia axyridis Pallas in laboratory and greenhouse. The results from this article provide a basis for utilizing H. axyridis in greenhouse during the winter. The experimenal design is proper and results are robust. There are some minor concerns.
1. Phtoperiod and illumination both used in this manuscript may mislead potential readers. Authors can select one of them to indicate the illumination time.
2. In the discussion, the author should also discuss the impact of extending illumination time on crop growth and reproduction.
Round 2
Reviewer 1 Report
Comments and Suggestions for Authors
The authors addressed all my comments. The manuscript was substantially improved. I think that it can be published